Paeoniflorin improves functional recovery through repressing neuroinflammation and facilitating neurogenesis in rat stroke model

Tang Hongli 1
Wu Leiruo 2
Chen Xixi 1
Li Huiting 1
Huang Baojun 1
Huang Zhenyang 1
Zheng Yiyang 1
Zhu Liqing zhuliqing@wzhospital.cn 3
Geng Wujun gengwujun@wzhospital.cn 1
1 Anesthesiology, the First Affiliated Hospital of Wenzhou Medical University , Wenzhou , Zhejiang , People’s Republic of China
2 Endoscopy Center, the First Affiliated Hospital of Wenzhou Medical University , Wenzhou , Zhejiang , People’s Republic of China
3 Clinical Laboratory, the First Affiliated Hospital of Wenzhou Medical University , Wenzhou , Zhejiang , People’s Republic of China
Maggi Laura
Electronic publication date: 2021 May 28
Publication date: 2021
Volume: 9
Electronic Location ID: e10921
Received 2020 Jul 16; Accepted 2021 Jan 19
Copyright: ©2021 Tang et al.
Copyright year: 2021
Copyright holder: Tang et al.
License: This is an open access article distributed under the terms of the Creative Commons Attribution License, which permits unrestricted use, distribution, reproduction and adaptation in any medium and for any purpose provided that it is properly attributed. For attribution, the original author(s), title, publication source (PeerJ) and either DOI or URL of the article must be cited.
License URL: https://creativecommons.org/licenses/by/4.0/

Keywords: Ischemic stroke, Paeoniflorin, Microglia, Neuroinflammation, JNK

Funding: Zhejiang public welfare technology research plan LGD20H290002 National Natural Science Foundation of China 81973620 81774109 Major Scientific and Technological Innovation Medical and Health projects of Wenzhou Science and Technology Bureau ZY2019015 This work was supported by the Zhejiang public welfare technology research plan (LGD20H290002), the National Natural Science Foundation of China (No. 81973620, No. 81774109), and Major Scientific and Technological Innovation Medical and Health projects of Wenzhou Science and Technology Bureau (ZY2019015). The funders had no role in study design, data collection and analysis, decision to publish, or preparation of the manuscript.

==============================
Background

Microglia, neuron, and vascular cells constitute a dynamic functional neurovascular unit, which exerts the crucial role in functional recovery after ischemic stroke. Paeoniflorin, the principal active component of Paeoniae Radix, has been verified to exhibit neuroprotective roles in cerebralischemic injury. However, the mechanisms underlying the regulatory function of Paeoniflorin on neurovascular unit after cerebral ischemia are still unclear.

Methods

In this study, adult male rats were treated with Paeoniflorin following transient middle cerebral artery occlusion (tMCAO), and then the functional behavioral tests (Foot-fault test and modified improved neurological function score, mNSS), microglial activation, neurogenesis and vasculogenesis were assessed.

Results

The current study showed that Paeoniflorin treatment exhibited a sensorimotor functional recovery as suggested via the Foot-fault test and the enhancement of spatial learning as suggested by the mNSS in rat stroke model. Paeoniflorin treatment repressed microglial cell proliferation and thus resulted in a significant decrease in proinflammatory cytokines IL-1β, IL-6 and TNF-α. Compared with control, Paeoniflorin administration facilitated von Willebrand factor (an endothelia cell marker) and doublecortin (a neuroblasts marker) expression, indicating that Paeoniflorin contributed to neurogenesis and vasculogenesis in rat stroke model. Mechanistically, we verified that Paeoniflorin repressed JNK and NF-κB signaling activation.

Conclusions

These results demonstrate that Paeoniflorin represses neuroinflammation and facilitates neurogenesis in rat stroke model and might be a potential drug for the therapy of ischemic stroke.

Introduction

Ischemic stroke is usually caused by disturbance of the blood supply to the brain, and it posts a significant threat for patients aged over 60 (Hankey, 2017; Roy-O’Reilly & McCullough, 2018). It is reported as one of the largest causes of disability and death in the world (Katan & Luft, 2018). Although the incidence, prevalence, and mortality of stroke are decreasing in the US and remain stable worldwide in the past decade (Guzik & Bushnell, 2017), it is still the primary cause of death and mortality in the past two decades in China, especially for patients over 60 years old (Feigin et al., 2014; Moraga & Collaborators GCoD, 2017; Wu et al., 2019b; Zhou et al., 2016b). The most common treatments are Intravenous alteplase (rtPA) and endovascular thrombectomy. Intravenous alteplase (rtPA) is the only FDA-approved drug for improving the disability of acute ischemic stroke patients. It is only effective when given within a very tight time window; otherwise, it may increase the risk of hemorrhagic transformation, and it does not affect mortality (Emberson et al., 2014; Strbian et al., 2014). Endovascular thrombectomy is only used as an additional method for rtPA treatment and it is only effective for 10% of ischemic stroke patients (Goyal et al., 2016). Furthermore, these treatments have a higher requirement for the local emergency unit on their ability to identify, transfer, and transport them to specialized stroke teams in a short time (Hankey, 2017). With the growth of the Chinese population and age, the number of stroke patients will increase significantly and leads to a great financial burden on both the Chinese public health system and families, and it is crucial to develop a new affordable and effective drug for ischemic stroke (Wu et al., 2019b).

Traditionally, microglia-mediated neuroinflammation is considered to play a major role in the pathophysiology of ischemic stroke since it is usually the first immune response step in the central nervous system (Moskowitz, Lo & Iadecola, 2010). Microglia are soon polarized into either inflammatory phenotype or anti-inflammatory phenotype in the peri-infarct region, based on the stimulus, the period, and the environment (Zong et al., 2019). They can either provoke damage or stimulate neuron repairs according to different pathophysiological conditions (Gordon & Taylor, 2005). However, it is reported that microglia activation could cause neurotoxic outcomes by increasing levels of inflammatory cytokines in the long term (Anttila et al., 2017). Multiple drugs, small molecules, and microRNA are reported in functional recovery of ischemic stroke by adjusting microglia polarization (Ma et al., 2017).

Paeoniflorin (PF), a monoterpene glucoside purified from the root of Paeonia lactiflora, is reported to have great potential to counter ischemic stroke as an anti-inflammatory and immunosuppressive agent (Xin et al., 2019). According to recent researches, paeoniflorin demonstrate its anti-inflammatory effects in multiple ways: it could reduce the inflammatory cytokine IL-17 and increase the anti-inflammatory cytokine IL-10 (Dai et al., 2015); it also could reduce the response of microglia to injury by suppressing IL-1β and TNF-α level (Zhou et al., 2016a); paeoniflorin also demonstrate its anti-inflammatory ability in multiple pathways by regulating different inflammation-related factors, for example, TNF-α, IL-1β, TGF-β1, Th-1 and etc. (Dai et al., 2015; Jin et al., 2011; Wang et al., 2012; Xin et al., 2019; Zhai & Guo, 2016). These findings not only illustrate the effects of paeoniflorin in multiple inflammatory disorders and neurodegenerative disease but also show the possibility of paeoniflorin using as an anti-inflammatory agent targeting at microglia of inflammatory phenotype in ischemic stroke. Some experiments on rats have exhibit paeoniflorin’s ability to protect the brain from ischemic damage through inhibiting different inflammatory responses meditated through microglia (Guo et al., 2012; Tang et al., 2010a; Zhang et al., 2015b). At present, we demonstrated that Paeoniflorin promotes functional recovery by repressing neuroinflammation and facilitating neurogenesis in rat stroke model.

Materials and Methods

MCAO model and Paeoniflorin delivery

Animal experiments were conduct under the ethic approval obtained from the Committee for Animal Experiments at Wenzhou Medical University (wydw2019-0903). Forty adult male Sprague-Dawley (SD) rats (6–8 weeks, 200–250 g, n = 10 of each group) were obtained from the Laboratory Animal Centre Wenzhou Medical University and kept under the specific-pathogen-free (SPF).

MCAO procedures were performed on all groups (the sham-operated group, vehicle experimental group, vehicle Paeoniflorin 5 mg/kg group and vehicle Paeoniflorin 10 mg/kg group) as previously described (Hata et al., 1998; Yu et al., 2018). In short, each rat received intraperitoneal anesthesia using choral hydrate (350 mg/kg) before exposing the correct vessels by a midline skin incision. The middle cerebral artery (MCA) was occluded by inserting a coagulated external carotid artery into the internal carotid artery. The MCA perfusion was allowed by withdrawing the suture after two hours. Verification of the occurrence of ischemia was examined by measuring the blood flow of MCA through a Laser Doppler flowmetry (LDF, Perimed, PeriFlux 5000). The same operations were performed on the sham-operated group rats exclude the MCA occlusion. All rats were kept on a heating pad (Malvern, UK) at 37.0 °C until the skin incision enclosure. After the previous procedure, the vehicle group was treated with (phosphate-buffered saline consists of 150 µl saline and 20% dimethyl sulfoxide [DMSO]) by intraperitoneal injection; while the paeoniflorin group was injected with Paeoniflorin (5 mg/kg and 10 mg/kg) twice per day for 14 days.

Animals were housed in a colony room under controlled temperature (22 °C), and a 12:12 h light-dark cycle, with food and water available. All rats were sacrificed by overetherization and decapitation. 

Behavioral tests

Neurological function assessment was performed by the mNSS test in accordance with the previous method (Chen et al., 2001). This test was carried out at day 0, 1, 3, 7 and 14 after MCAO. This test, including the measurements for the motor, sensory, beam balance, and reflexes (Schaar, Brenneman & Savitz, 2010). This test was graded on a scale of 0–18. More serious injuries usually lead to higher scores.

The sensorimotor function was evaluated via the foot-fault test as the method reported by Zhang et al. (2015a). The number of foot-fault errors, paw slipping numbers, were recorded when the rat is trained to cross an elevated grid at day 0, 1, 3, 7 and 14 after MCAO.

Cell culture

The murine BV2 microglial cells were procured from the ATCC (Manassas, VA, USA), and then maintained in DMEM with 10% FBS, 1% penicillin, and streptomycin (Sigma), and were cultured at 37 °C in an incubator with 5% CO2. The oxygen-glucose deprivation (OGD) was performed by exposing BV2 cells to DMEM without glucose or serum and incubating in a specific environment (5% CO2 and 95% N2) for 6 h. BV2 microglial cells were then incubated in Paeoniflorin (10 µM) for 6 h before TNF-α, IL-1β, IL-6 levels were measured by qPCR analysis.

Quantitative real-time PCR (qPCR)

The total RNA was isolated from BV2 cells or OGD model cells in the absence or presence of 10 µM Paeoniflorin treatment for 6 h with Trizol kit (Solarbio, Beijing, China) following the manufacturer’s manual. Reverse transcription (RT) was implemented by Prime Script™ Master Mix (Katara, Japan). qPCR was performed with 2 × SYBR Green Mix (Solarbio) on an Applied Biosystems 7500 Fast Real-Time PCR Systems (Thermo Fisher Scientific, MA). β-actin was chosen as an internal reference and the fold changes were calculated using the 2−ΔΔCt method.

Western blotting

Total protein was purified from the brain tissues of 14 days after MCAO in the absence or presence of 10 mg/kg Paeoniflorin treatment using lysis buffer and then centrifuged at 12,000 rpm for about 15 min. The protein concentration was measured with the BCA kit (Pierce, IL). The western blotting was carried out as the method reported by Zhu et al. (2019). The cell was cultured with primary antibodies, anti-Iba-1 antibody (1 µg/ml; ab5076; Abcam, MA), anti-JNK antibody (1:1000; ab179461; Abcam, MA), anti-phosphorylated-JNK antibody (1:1,000; #9251; Cell Signaling Technology), anti-P65 antibody (0.5 µg/ml; ab16502; Abcam), anti-Histone H3 antibody (1:1,000; ab215728; Abcam) and anti-actin antibody (1:1000, ab6276, Abcam), at 4 °C overnight. After washing, horseradish peroxidase (HRP)-conjugated secondary antibodies were incubated with membranes for 1 h.

(Enzyme linked immunosorbent assay) ELISA

The ELISA assay was carried out to detect TNF-α, IL-1β and IL-6 content using ELISA kits (PT516/PI303/PI328; Beyotime, China) following the manufacture’s manual. In short, the brain tissues were extracted at 14 days after MCAO in the absence or presence of 10 mg/kg Paeoniflorin treatment, and then the supernatant of the brain tissue homogenate (1:20 dilution) was added to 96-well plates that are coated with indicated antibodies, in order to quantify the protein level of TNF-α, IL-1β, IL-6 in the tissue of the brain. After the reaction was completed, the absorbances of the sample were measured at 450 nm with a microplate reader. At least three repeated measurements were taken for each sample.

Immunofluorescence staining

The immunofluorescence staining for brain tissues was carried out as the method reported by Burton (Burton, Sparkman & Johnson, 2011), and the brain tissues were extracted at 14 days after MCAO in the absence or presence of 10 mg/kg Paeoniflorin treatment. The cell was cultured with specific primary antibody against Iba-1 (1:1,500; ab178846) at 4 °C overnight, and then incubated with Anti-rat Alexa Fluor 488 (1:1000; CST) as a secondary antibody. The samples were evaluated with the LEICA TCS SPE microscope (Leica, Germany).

Statistics

Data from the experiments were presented as mean ± standard deviation (SD). The difference between two groups was compared using two-tailed  student’s t-test, or one-way analysis of variance (ANOVA)  followed by the Scheffé test. Values less than 0.05 was considered significant differences.

Results

Paeoniflorin improved the functional recovery in rat stroke model

The role of Paeoniflorin in functional recovery following cerebral ischemic injury was first tested in rat stroke model. The rats were subjected to MCAO and then Paeoniflorin was intraperitoneally administrated for 14 days at 2 h after MCAO. The Foot-fault test and mNSS were performed at day 0, 1, 3, 7 and 14 after MCAO. There were no remarkable differences in neurological functional deficits at day 1 after Paeoniflorin administration, while Paeoniflorin markedly improved functional recovery at day 3, 7 and 14 following MCAO, compared with vehicle experimental group (Fig. 1A). Meanwhile, Paeoniflorin administration significantly reduced the frequency of forelimb and foot faults at day 3, 7 and 14 following MCAO (Fig. 1B). Importantly, better protective effects were obtained with higher Paeoniflorin concentrations in rat stroke model (Figs. 1A and 1B). The current data verified that Paeoniflorin enhanced the functional recovery in rat stroke model.

Figure 1 Paeoniflorin improved the functional recovery in rat stroke model.

(A) Theneurological function of rats was measured by mNSS at day 0, 1, 3, 7 and 14 after MCAO and intraperitoneal injection of vehicle or Paeoniflorin (5 mg/kg and 10 mg/kg; n = 10 of each group). Data was analyzed by one-way ANOVA followed by the Scheffé test. n = 10. **P < 0.01 vs. control group. (B) A foot-fault test was performed at day 0, 1, 3, 7 and 14 after MCAO and intraperitoneal injection of vehicle or Paeoniflorin (5 mg/kg and 10 mg/kg). Data was analyzed by one-way ANOVA followed by the Scheffétest. n = 10. **P < 0.01 vs. control group. mNSS, Modified neurological severity scores; MCAO, middle cerebral artery occlusion.

Paeoniflorin repressed microglia activation in rat stroke model

Microglia are the resident innate immune cells in the brain. Microglia in resting state are responsible for routine immune surveillance (Guruswamy & ElAli, 2017). After cerebral ischemic injury, microglia are activated in morphological and phenotypical changes (Jayaraj et al., 2019), and the severity of cerebral ischemic injury is correlated with the activated state of microglia (Emmrich et al., 2015). To explore whether Paeoniflorin repressed the activation of microglia in rat stroke model, microglia were analyzed using immunofluorescence analysis after Paeoniflorin administration. As shown in Fig. 2A, rats subjected to MCAO showed a significant microglial proliferation in ipsilateral cortex, whereas additional Paeoniflorin treatment repressed MCAO-induced microglial proliferation. Moreover, Paeoniflorin inhibited MCAO-induced microglial activation as measured by the intensive ramified Iba-1-positive staining (Fig. 2A). To further assess the function of Paeoniflorin on microglial activation, the protein expression of Iba-1 in ipsilateral cortex was analyzed. Rats subjected to MCAO showed a significant increase of Iba-1 protein level in ipsilateral cortex, whereas Paeoniflorin treatment inhibited this increase compared with the control (Figs. 2B and 2C). Based on this result, the role of Paeoniflorin in regulating the secretion of pro-inflammatory cytokines was assessed in cerebral tissues. Figures 3A–3C showed that rats subjected to MCAO showed a significant increase in the production of IL-1β, TNF-α and IL-6, whereas Paeoniflorin administration repressed pro-inflammatory cytokines production in cerebral tissues. These results demonstrated that Paeoniflorin exerts the important role in repressing microglial activation and subsequent neuroinflammation in rat stroke model.

Figure 2 Paeoniflorin repressed microglia activation in rat stroke model.

(A) The activation of microglia was analyzed using immunofluorescence analysis in ipsilateral cortex after MCAO and intraperitoneal injection of vehicle or Paeoniflorin (10 mg/kg) n = 3. Scale bar, 20 µm (B and C) Western blot and quantitative analysis of the protein expression level of Iba-1 in ipsilateral cortex after MCAO and intraperitoneal injection of vehicle or Paeoniflorin. *P < 0.05, **P < 0.01. n = 3. MCAO, middle cerebral artery occlusion; Iba-1, Ionized calcium bindingadaptor molecule-1.

Figure 3 Paeoniflorin inhibits the secretion of pro-inflammatory cytokines in rat stroke model.

(A–C) The levels of pro-inflammatory cytokines TNF-a (A), IL-1β (B) and IL-6 (C) were determined by ELISA assay using ELISA kits in brain tissues after MCAO and intraperitoneal injection of vehicle or Paeoniflorin (10 mg/kg). n = 3. *P < 0.05, **P < 0.01. IL-1β, Interleukin-1 beta; TNF, tumor necrosis factor; IL-6, Interleukin-6; ELISA, enzyme linked immunosorbent assay; MCAO, middle cerebral artery occlusion.

Paeoniflorin repressed microglial viability and inflammatory cytokines production in vitro

To test the role of Paeoniflorin in microglial activation in vitro, the viability of murine BV-2 microglial cells on oxygen and glucose deprivation (OGD) was assayed in the presence or absence of Paeoniflorin. Figure 4A showed that the viability of BV-2 cells was reduced after Paeoniflorin treatment under normoxia or OGD, indicating that Paeoniflorin inhibited BV-2 cells activity. The effect of Paeoniflorin on pro-inflammatory cytokines production in vitro was further assessed. Here OGD model was used as an in vitro model of ischemia. The mRNA expression level of IL-1β, TNF-α and IL-6 was assayed by Real time-PCR under OGD in the presence or absence of Paeoniflorin, and 18s was used as internal gene to control the different amount of RNA input due to the different cell viability. Figures 4B–4D showed that IL-1β, TNF-α and IL-6 expression level was upregulated in BV-2 cells under OGD, but additional treatment of Paeoniflorin repressed OGD-induced increase in IL-1β, TNF-α and IL-6 expression. These data suggested that Paeoniflorin improved functional recovery, at least in part, by suppressing microglia-mediated inflammatory response.

Figure 4 Paeoniflorin repressed microglial viability and inflammatory cytokines production in vitro.

(A) BV2 microglia viability was determined by CCK-8 assay using CCK-8 kit in BV2 cells with vehicle or Paeoniflorin treatment (10 µM) and OGD models (BV2 cells subjected to oxygen and glucose deprivation) treated with vehicle or Paeoniflorin treatment (10 µM) for 24 h. n = 3. *P < 0.05. (B–D) The mRNA expression levels of IL-1β (B), TNF-α (C), and IL-6 (D) were detected by qPCR analysis in BV2 cells with vehicle or Paeoniflorin treatment (10 µM) and OGD models treated with vehicle or Paeoniflorin treatment (10 µM) for 24 h. n = 3. **P < 0.01. CCK-8, Cell Counting Kit-8; OGD, oxygen-glucose deprivation; IL-1β, Interleukin-1 beta; TNF, tumor necrosis factor; IL-6, Interleukin-6; qPCR, Quantitative Real-time PCR.

Paeoniflorin facilitated neurogenesis and vasculogenesis in rat stroke model

The marker of endothelial cells of cerebral blood vessels, vWF, was next assessed after Paeoniflorin administration. As shown in Figs. 5A and 5B, Paeoniflorin treatment resulted in a significant upregulation of the vWF expression in peri-infarct zone compared with control in rat stroke model. The indication of migrating neuroblasts, doublecortin, was also assessed in peri-infarct zone after Paeoniflorin treatment. Figures 5C and 5D showed that Paeoniflorin treatment resulted in a significant increase of the doublecortin expression in peri-infarct zone compared with control. Furthermore, Paeoniflorin reduced the TUNEL-positive neurons compared with control (Figs. 5E and 5F). These results demonstrated that Paeoniflorin promoted neurogenesis and vasculogenesis after ischemic stroke.

Figure 5 Paeoniflorin facilitated neurogenesis and vasculogenesis in rat stroke model.

(A and B) The marker of endothelial cells of cerebral blood vessels (vWF) expression was assessed in peri-infarct zone compared with control of rat stroke model after Paeoniflorin administration by Immunofluorescence. n = 3. *P < 0.05. (C and D) The indication of migrating neuroblasts (doublecortin) expression was assessed in peri-infarct zone compared with control of rat stroke model after Paeoniflorin administration by Immunofluorescence. **P < 0.01. (E and F) The effects of paeoniflorin on apoptotic cell was verified by TUNEL staining in rat stroke model treated with or without Paeoniflorin. Positive expression in TUNEL staining was exhibited. Rats treated with saline and PF (10 mg/kg) for 14 days after MCAO; n = 5. Scale bar, 50 µm. **P < 0.01. TUNEL, terminal dexynucleotidyl transferase (TdT)-mediated dUTP nick end labeling.

Paeoniflorin repressed JNK and NK-κB signaling activation

Given the regulatory role of Paeoniflorin in Jun N-terminal kinase (JNK) signaling in many diseases (Liu et al., 2019; Wu et al., 2019a; Zhang et al., 2017a) and the effect of JNK on regulating inflammatory response (Chen et al., 2018), we next investigated whether Paeoniflorin repressed JNK signaling activation in rat stroke model. The results from western blot analysis showed that the level of p-JNK (phosphorylated JNK) was markedly upregulated in cerebral tissues at 4 h after MCAO, whereas additional treatment of Paeoniflorin weakened the p-JNK level in cerebral tissues compared with control, suggesting that ischemia-induced activation of JNK signaling was repressed by Paeoniflorin administration (Figs. 6A and 6B).

Figure 6 Paeoniflorin repressed JNK and NK-κB signaling activation.

(A and B) Western blot and quantitative analysis of the protein expression level of JNK and phosphorylated JNK (p-JNK) after MCAO and intraperitoneal injection of vehicle or Paeoniflorin. n = 3. **P < 0.01, # P < 0.05. (C and D) Western blot and quantitative analysis of the protein expression level of nuclear p65 and Histone H3 after MCAO and intraperitoneal injection of vehicle or Paeoniflorin. (E) Nuclear translocation of p65 was indicated using an immunofluorescence assay after MCAO and intraperitoneal injection of vehicle or Paeoniflorin. n = 3. *P < 0.05. Scale bar, 20 µm. MCAO, middle cerebral artery occlusion.

It is well-known that JNK could activate NF-κB signaling to aggravate the progression of inflammatory response (Pan et al., 2013; Zheng et al., 2020). We thus tested whether neuroinflammation-relieving role of Paeoniflorin was associated with NF-κB signaling activation. Figures 6C and 6D showed that the nuclear protein level of p65 was markedly upregulated in rat stroke model, whereas additional Paeoniflorin treatment weakened the nuclear p65 level in cerebral tissues compared with control.

Immunofluorescence analysis also showed that Paeoniflorin administration partially inhibited MCAO-induced nuclear translocation of NF-κB p65 in primary microglia (Fig. 6E). Taken together, the current results demonstrated that Paeoniflorin improves functional recovery through repressing neuroinflammation and facilitating neurogenesis by inhibiting JNK- NF-κB signaling in rat stroke model.

Discussion

In the present study we investigated the potential role of Paeoniflorin in protecting against cerebral ischemia injury, and identified the effect of Paeoniflorin on regulating microglia, neuron, and vascular cells in rat stroke model. The present data verified that (i) Paeoniflorin improved the functional recovery in rat stroke model, (ii) Paeoniflorin repressed microglia activation in rat stroke model, (iii) Paeoniflorin repressed microglial viability and inflammatory cytokines production in vitro, (iv) Paeoniflorin facilitated neurogenesis and vasculogenesis in rat stroke model, (v) Paeoniflorin repressed JNK and NK-κB signaling activation. These data verify the important role of Paeoniflorin in repressing neuroinflammation and facilitating neurogenesis, and may provide a new opportunity to the therapy of ischemic stroke.

Paeoniflorin is the primary active component extracted from the roots of Paeonia plants. Over the past decade, the protective role of Paeoniflorin in central nervous system has been verified (She et al., 2019). Paeoniflorin can rapidly pass through the BBB (blood–brain barrier) and repress microglia-mediated inflammatory response in the central nervous system (Liu et al., 2006). Our previous study demonstrated that Paeoniflorin administration enhances motor function recovery and reduces the histopathological damage in a rat model of spinal cord injury (Wang et al., 2018). Paeoniflorin increases neuronal survival and represses neuroinflammation by inhibiting NF-κB signaling (Wang et al., 2018).

The role of Paeoniflorin in ischemic stroke has also been identified. Gao et al. demonstrated that Paeoniflorin administration represses MCAO-induced over-activation of microglia and astrocytes, and thus inhibits the production of pro-inflamamtory cytokines including IL-1β, TNFα, COX2 and iNOS (Guo et al., 2012). They further verified that Paeoniflorin exerts a protective role against ischemic injury by inhibiting of JNK, p38 and NF-κB signaling activation (Guo et al., 2012). Ko et al. (2018) demonstrated that Paeoniflorin decreases neurological deficit score and enhances motor function. Tang et al. (2010b) suggests that both pre-treatment and post-treatment with PF reduced the ratio of cerebral infarction area; pre-treatment with Paeoniflorin also reduced the neurological deficit score. Zhang et al. (2017b) indicates that the protective effect of Paeoniflorin on cerebral ischemia reperfusion injury is possible through regulating the Ca2+/CaMKII/CREB signaling pathway. These studies clearly showed the potential role of Paeoniflorin in alleviating ischemic cerebral injury. However, the role and underlying mechanism of paeoniflorin in the neurogenesis of cerebral ischemia-reperfusion injury are still unclear. Moreover, the current data demonstrated that Paeoniflorin improves functional recovery through repressing neuroinflammation and facilitating neurogenesis in rat stroke model. These data further enhance the persuasiveness of Paeoniflorin in treating ischemic stroke.

Pro-inflammatory cytokines contribute to stroke-related brain injury (Liang et al., 2011). During ischemia, cytokines, such as IL-1β, IL-6 and TNF-α are produced by a variety of activated cell types, including microglia and neurons (Huang, Upadhyay & Tamargo, 2006). Our results document that Paeoniflorin treatment contributes to inhibit microglial cell proliferation and thus results in a significant decrease in proinflammatory cytokines. However, the inhibitory effect of Paeoniflorin treatment on microglia proliferation is due to cytotoxicity or regulation of other signaling pathways remains to be fully elucidated. How does Paeoniflorin regulate the expression of these inflammatory cytokine remains still unclear. Guo et al. (2012) demonstrates that Paeoniflorin produces a delayed protection in the ischemia-injured rats via inhibiting MAPKs/NF-κB mediated peripheral and cerebral inflammatory response. We found that ischemia-induced activation of JNK signaling was repressed by Paeoniflorin administration. It is well-known that JNK could activate NF-κB signaling to aggravate the progression of inflammatory response (Pan et al., 2013; Zheng et al., 2020). These data suggest that Paeoniflorin regulate these inflammatory cytokine by suppressing JNK- NF-κB signaling pathway in rat stroke model.

In the current study, Paeoniflorin administration promotes von Willebrand factor (an endothelia cell marker) and doublecortin (a neuroblasts marker) expression compared with control, suggesting that Paeoniflorin facilitates neurogenesis and vasculogenesis in rat stroke model. the underlying mechanism of Paeoniflorin facilitates neurogenesis still unclear. It has been reported that Microglia and proinflammatory cytokines can enhance neuroprotection or neurotoxicity during the inflammatory response, so they are a double-edged sword in neurogenesis (Ekdahl, Kokaia & Lindvall, 2009). Ekdahl, Kokaia & Lindvall (2009) reported that the number of activated microglia was negatively correlated with the survival rate of neonatal hippocampal neurons. Cai et al. (2019) Arg-DG attenuates LPS-impaired neurogenesis and suppresses the production of LPS-induced pro-inflammatory cytokines in hippocampal DG by blocking microglial activation. Our data suggests Paeoniflorin treatment contributes to inhibit microglial cell proliferation and thus results in a significant decrease in proinflammatory cytokines. Based on these data. we concluded that Paeoniflorin inhibits neuroinflammation and facilitates neurogenesis by suppressing the production of inflammatory cytokine and activation of microglia.

In conclusion, we showed that Paeoniflorin exhibits a sensorimotor functional recovery. Paeoniflorin treatment contributes to inhibit microglial cell proliferation and thus results in a significant decrease in proinflammatory cytokines. Meanwhile, Furthermore, we demonstrated that Paeoniflorin represses JNK and NF-κB signaling activation. The current data verify that Paeoniflorin inhibits neuroinflammation and facilitates neurogenesis in rat stroke model and might be a potential drug for the therapy of ischemic stroke.

Supplemental Information

Data S1 Raw Data

Western Blot, immunofluorescence, and qPCR data.

Click here for additional data file.

Additional Information and Declarations

Competing Interests

Author Contributions

Animal Ethics

Data Availability

The authors declare there are no competing interests.

Hongli Tang, Leiruo Wu and Xixi Chen conceived and designed the experiments, prepared figures and/or tables, and approved the final draft.

Huiting Li and Baojun Huang performed the experiments, prepared figures and/or tables, and approved the final draft.

Zhenyang Huang performed the experiments, authored or reviewed drafts of the paper, and approved the final draft.

Yiyang Zheng, Liqing Zhu and Wujun Geng analyzed the data, authored or reviewed drafts of the paper, and approved the final draft.

The following information was supplied relating to ethical approvals (i.e., approving body and any reference numbers):

The Laboratory Animal Ethics Committee of Wenzhou Medical University approved this research (wydw2019-0903).

The following information was supplied regarding data availability:

Raw data are available in a Supplemental File.

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
