# Peer review of "Paeoniflorin improves functional recovery through repressing neuroinflammation and facilitating neurogenesis in rat stroke model"

_PeerJ, doi:10.7717/peerj.10921_

## Round 0.1 · original submission · Major Revisions

The paper should be strongly revised, according to the reviewer's comments. In particular, data analysis, experimental details, introduction and discussion paragraph should be implemented.

·

Basic reporting

In this paper the authors reported that Paeoniflorin represses neuroinflammation and facilitates neurogenesis in rat stroke model and might be a potential drug for the therapy of ischemic stroke.
In particular they reported that Paeoniflorin improved the functional recovery, facilitated neurogenesis and vasculogenesis, repressed microglia activation, repressed JNK and NK-κB signaling activation in rat stroke model and repressed microglial viability and inflammatory cytokines production in vitro.

However, these data are not new since there are numerous papers in the literature that, using exactly the same animal model, demonstrated that Paeoniflorin is neuroprotective against ischemia-induced brain damage by inhibiting the inflammatory response mediated by NF-KB. (See Guo et al., 2012; Zhi Liu et al., 2005; Xiao et al 2005; Tang et al., 2010; Zhang et al., 2015; Zhang et al., 2017; Wu et al., 2020 etc etc).
The only new data is that concerning neurogenesis and vasculogenesis.

Moreover, the analysis of the data is also incorrect since the expression of the various markers observed in the ipsi-lateral hemisphere should be normalized for the same analysis conducted in the contra-lateral hemisphere as internal control.

Experimental design

The Methods are not described with sufficient detail & information to replicate;
for examples it is not specified how long after the ischemic damage the analyzes were conducted. The authors refer to timing only with respect to neurological tests.

The analysis of most of the data is incorrect:
1) the expression of the various markers observed in the ipsi-lateral hemisphere should be normalized for the same analysis conducted in the contra-lateral hemisphere as internal control.
2) the reduction of IL1b, TNFa, IL6 induced by Paeoniflorin in BV2 cells (Fig 4B-D) should be normalized for the number of cells since they observed a reduction of BV2 viability after Paeoniflorin treatment (Fig4A). Moreover they cannot exclude that Paeoniflorin might be toxic for BV2 cells so it is not correct to say that Paeoniflorin inhibits BV2 activity.

Validity of the findings

The data reported are not new since there are numerous papers in the literature that, using exactly the same animal model, demonstrated that Paeoniflorin is neuroprotective against ischemia-induced brain damage by inhibiting the inflammatory response mediated by NF-KB. (See Guo et al., 2012; Zhi Liu et al., 2005; Xiao et al 2005; Tang et al., 2010; Zhang et al., 2015; Zhang et al., 2017; Wu et al., 2020 etc etc).
The only new data is that concerning neurogenesis and vasculogenesis.

Additional comments

No comment

Reviewer 2 ·

Basic reporting

The article is written in a clear English.
The introduction needs more details, since there is not any information regarding the interplay between stroke and neurogenesis, neither between inflammation, in particular at central nervous system level, and neurogenesis. The authors should mentioned the potential effects of stroke and polaridez microglia on neurogenesis, and the background according to whom they decided to explore the level of adult neurogenesis in their experimental model.

The paper has an acceptable format, however the figures should be improved. In particular, the details of the scale bar used should be insert in Figure 2 and 5.
In the text it has been reported that behavioral tests have been performed with 5 animals per groups (which is a very small sample for behavioral assessment),while in the raw data file on the first and second page have been reported the results of 3 animals per groups. Which is the correct number?

Did the author run a power analysis to calculate the minimum sample size required for such experimental design?

Experimental design

The behavioral data have been obtained with animals treated with two doses (5mg/kg and 10 mg/kg) and both were effective following 3 days of treatment, however the other analyses were performed comparing only the vehicle with the higher dose, could the authors explain such decision? In addition, for the behavioral results should be rub an ANOVA analysis since the data consist in repetitive measures between subjects.

Part of the analyses were not performed on the sham group, could the authors provide such data or explains such decision?

The authors should specify the number of subjects per group used for the immunofluorescence and molecular analyses.

The authors should provide more information regarding the timeline of the experiments; in particular, it is not clear in which timepoint the molecular analyses (e.g. ELISA and immunofluorescence staining) were performed.

Validity of the findings

Although the aim and the impact of this work have been appropriately stated, discussion and conclusions should be improved adding the potential molecular and cellular mechanisms underlying the treatment effects. Again, should be implemented the interplay between stroke consequences, inflammation and neurogenesis levels, and the therapeutic action of the treatment.

---

## Round 0.2 · Minor Revisions

The reviewers have previously raised important criticisms which they described in detail. The authors have revised many parts of the manuscript that have been positively re-evaluated. Nevertheless, in this form, the manuscript could be accepted only if additional experiments can be provided, in particular on BV2 cells, as previously suggested by the reviewer. The author's response to this part is not clear enough and need more specific methodologies.

Reviewer 2 ·

Basic reporting

No comment

Experimental design

No comment

Validity of the findings

no comment

---

## Round 0.3 · Minor Revisions

The reviewer discussed the possibility to normalize the results on BV2 cells on the total amount of cells. Nevertheless, the authors at least need to better clarify their methods and interpretation on Paeoniflorin's role in repressing microglial viability, activation and inflammatory cytokines production.

---

## Round 0.4 · accepted · Accept

In this form, I consider the paper suitable for publication on PeerJ.